# Metabolomics and Physiological Approach to Understand Allelopathic Effect of Horseradish Extract on Onion Root and Lettuce Seed as Model Organism

**DOI:** 10.3390/plants10101992

**Published:** 2021-09-23

**Authors:** Tyler Simpson, Kang-Mo Ku

**Affiliations:** 1Division of Plant and Soil Sciences, West Virginia University, Morgantown, WV 26505, USA; tjsimpson@mix.wvu.edu; 2Department of Horticulture, College of Agriculture and Life Sciences, Chonnam National University, Gwangju 61886, Korea; 3BK21 Interdisciplinary Program in IT-Bio Convergence System, Chonnam National University, Gwangju 61186, Korea

**Keywords:** *Armoracia rusticana*, allelopathy, *Allium cepa* L., cell division, ROS, *Lactuca sativa*

## Abstract

In the present study, we assessed the allelopathic effects of various concentrations (0%, 0.1%, 0.2%, and 0.3%) of horseradish root extract (HRE) on onion root. The average growth of onion root tips during the 0% HRE treatment (deionized water treatment) was 0.9 cm/day, which was the highest among the growth rates obtained with all HRE treatments. Moreover, the average growth during 0.3% HRE treatment was 0.1 cm/day. During cell cycle analysis, the mitotic phase fraction of the control (deionized water treatment) cells was 6.5% of all dividing cells, with this percentage being the highest among the values obtained for all treatment groups. In the control group, all cell cycle phases were identified; however, in the 0.1%, 0.2%, and 0.3% treatment groups, telophase was not identified. The ROS accumulation area of the onion root decreased, as the HRE treatment concentration increased. In the control root, the area of dead tissue was 0%; however, in the 0.1% and 0.2% HRE treatment roots, the ratio was 5% and 50%, respectively. These findings indicate that the allelopathic effect of HRE depends on the concentration of HRE applied to the onion root.

## 1. Introduction

Horseradish (*Armoracia rusticana* G. Gaertn., B. Mey, and Scherb) is a root crop belonging to the Brassicaceae family, which also includes several vegetables such as mustard, cabbage, broccoli, Brussels sprouts, and kale. It is most commonly used as a condiment due to its unique pungent flavor. The intense flavor of horseradish is primarily due to the hydrolysis products of glucosinolates, which are sulfur-containing secondary plant metabolites [1]. Sinigrin is the most dominant glucosinolate in horseradish, accounting for 83% of the total glucosinolate content in the root tissue and 92% in the leaves [2]. When a tissue is disrupted, sinigrin reacts with an enzyme myrosinase, which is compartmentalized in other cells. Hydrolysis products allyl isothyocyanate (AITC), allyl cyanide, 1-cyano-2,3-epithiopropane, and allyl thiocyanate are produced by this reaction [3]. These hydrolysis products act as a defense mechanism in plants against fungi, insect herbivores, and other abiotic stresses including drought and ultraviolet (UV) [1]. Moreover, glucosinolate hydrolysis products have potential uses as medical and agricultural agents. These compounds are known to exhibit anticarcinogenic effects against bladder, breast, colon, lung, stomach, and liver cancers in animal models [4]. Furthermore, they can also be used as cover crops and green alternatives to herbicides [5].

Glucosinolates and their hydrolysis products are crucial in allelopathy [6], which is defined as the chemical stimulation or depression of a plant or microorganism either directly or indirectly via secondary metabolites produced by a distinctly separate plant or microorganism [7]. Additionally, the allelopathic effects of various plant extracts and phytochemicals have been reported in a previous study [8]. Black mustard (*Brassica juncea*) extract was tested for its effects on the seed germination and cell cycle of pea [9] as well as on the germination and radical growth of radish [10], and it was found that the allelopathic effect of black mustard was related to AITC. Åsberg et al. [11] reported that AITC reduced the mitotic phase during the cell cycle in *Arabidopsis*. Zhao et al. reported ABA inhibition of the inward K^+^ channels of guard cells and the stomatal openings of *Arabidopsis* treated with AITC [12]. Hara et al. [13] reported that AITC can be harmful at high concentrations and may enhance the glutathione-s-transferase (GST) activity at low concentrations in *Arabidopsis*. AITC’s effects have primarily been researched in human and animal cells due to its potential anticarcinogenic properties. Moreover, AITC promoted apoptosis, arrested the cell cycle in a cell line-dependent manner, increased cellular oxidative stress, and decreased the levels of glutathione and other thiols in cells [14,15,16]. Although some functions and allelopathic effects of AITC on animal’s cells have been witnessed, the mechanism underlying the allelopathic effect on plants at the cellular level has not yet been fully elucidated. 

In the present study, we examined the allelopathic effects of horseradish extract (HRE), including the effects of glucosinolate and its derivatives, at the cellular level on onion (*Allium cepa*) root growth. It is relatively easier to observe the cell cycle of the cells in the onion root because of the large chromosome size, and the meristematic cells situated in the root tip serve to be very suitable for studies on cell cycle and development [17,18]. Lettuce (*Lactuca*
*sativa*) was used as another model organism in this study due to its high germination rate and relatively uniform biological replications [19]. We tried to clarify the effects of the HRE extract on plant cells using the onion by measuring the root length, observing the cell cycle, and evaluating the degree of reactive oxygen species (ROS) accumulation as well as metabolite changes during lettuce seed germination.

## 2. Results and Discussion

### 2.1. Onion Root Growth after Treatment with Various Concentrations of HRE

Allelopathic activity varies depending on the concentration of the extract, target species, and plant tissues used to extract the chemicals [20]. Our results confirm the allelopathic effect of HRE on onion root and shows that this effect was dependent on the concentration of HRE.

Figure 1 illustrates the average growth per day of onion root tips over 3 days following treatment with various concentrations of HRE (0%, 0.1%, 0.2%, and 0.3%). The average growth of onion root tips during the 0% HRE treatment (deionized water treatment) was 0.94 cm/day, which was the highest among the growth rates following all HRE treatments. The average growth of onion root tips during 0.1% and 0.2% HRE treatments was 0.48 and 0.36 cm/day, respectively, and differed significantly from the growth rate obtained with the 0% HRE treatment (*p* < 0.05). Moreover, the average growth with the 0.3% HRE treatment was 0.06 cm/day and differed significantly from that obtained with the 0% HRE treatment (*p* < 0.01).

Jafariehyazdi et al. [21] assessed the allelopathic effect of *Brassica napus*, *Brassica rapa*, and *Brassica juncea* on the germination of sunflower seed. Jafariehyazdi et al. [21] reported that all aqueous extracts significantly affected sunflower germination and the root growth of sprouts compared to water control. With higher concentrations of the extract, strong growth inhibition was observed, which is consistent with the results of the present study.

### 2.2. Onion Root Cell Mitotic Stage following Treatment with Various Concentrations of HRE

The results of the cell cycle analysis of the onion root are illustrated in Figure 2. Figure 2A depicts the percentages of the cells in the various stages of mitosis 3 days after treatment with various HRE concentrations. In the control root (deionized water treatment), 6.5% of the cells were in the mitotic phase, with this percentage being the highest among all values obtained for all concentrations of HRE treatment. Moreover, 4%, 4.3%, and 3.2% of the cells were in the mitotic phase after treatment with 0.1%, 0.2%, and 0.3% HRE, respectively. Prophase constituted the largest percentage of all cell cycle phases following all HRE treatments. 

Furthermore, Figure 2A depicts the percentages of each mitotic phase to total dividing cells. In the control group, all cell cycle phases, namely, the prophase, metaphase, anaphase, and telophase were identified; however, after treatment with 0.1%, 0.2%, and 0.3% HRE, telophase was not identified. The percentage of the cells in the prophase increased with an increase in the HRE concentration and constituted the highest percentage of all cell cycle phases following 0.3% HRE treatment; moreover, this percentage significantly differed from those of other treatments (*p* = 0.05). The percentages of metaphase and anaphase decreased with increasing HRE concentrations. The percentages of metaphase and anaphase were lowest following 0.3% HRE treatment and significantly differed from those obtained with other treatments (*p* = 0.05). 

In this study, we observed that onion root tips treated with HRE were shorter than the control, regardless of the treatment concentration. Åsberg, Bones, and Øverby [11] reported similar results, which suggested that *A. thaliana* cells treated with AITC were smaller, on average, than control cells. We, thus, hypothesized that HRE would affect cell development. When onion root cells were treated with HRE, the late phases of mitosis were almost entirely absent, especially anaphase and telophase. Cells attempting to progress through the later phases of mitosis exhibited clumping of chromatids, showing an inability to be pulled apart. This mechanism relies on microtubules, and Øverby et al. [22] showed that AITC disintegrated microtubules in *A. thaliana*. Thus, HRE inhibited mitosis in onion cells, which likely reduced growth of onion cells, resulting in short root tips.

### 2.3. Onion Microscopic Characteristics during Cell Division following Treatment with Various Concentrations of HRE

The microscopic images of the mitotic stages after treating the onion root with various HRE concentrations are illustrated in Figure 3. In the control root tip, all cell cycle stages were clearly observed. Chromosomes appeared blurred as the HRE treatment concentration increased. The stage of anaphase was not observed in the 0.3% HRE treatment group, and the stage of telophase was not observed in the 0.1%, 0.2%, and 0.3% HRE treatment groups. The cells of the 0.1%, 0.2%, and 0.3% HRE treatment groups exhibited chromatids that showed an inability to separate and move to the opposing poles. This mechanism relies on microtubules, and this finding is consistent with that of by Øverby et al. [22], who reported an AITC-induced disintegration of microtubules in *A. thaliana*.

Several studies have reported that ITCs induce cell cycle arrest in cancer cells, which is often observed upon treatment with microtubule inhibitors [11,23,24]. Especially, Mi et al. [24] reported that the isothiocyanates including benzyl isothiocyanate, phenethyl isothiocyanate, and sulforaphane covalently bind with cysteine in the tubulin of A549 human cancer cells.

In Figure 3, we show that glucosinolate not only inhibits cell division, but also affects the statue of chromosomes. Musk et al. [25] investigated the genotoxicity and cytotoxicity of allyl and phenethyl isothiocyanates (AITC and PEITC, respectively) derived from sinigrin and gluconasturtiin. According to their study, AITC treatment did not reveal chromosome aberrations or sister chromatid exchanges (SCEs) at 1.0 µg/mL concentration, PEITC presented aberrations and SCEs at 0.9 µg/mL, and the other glucosinolates-induced SCEs at 2 mg/mL. Collectively, ITC revealed high genotoxicity at a relatively lower concentration than glucosinolate and had a remarkable cytotoxic effect compared to that of the glucosinolates. The effects of HRE on the cytoplasmic components and its signals as well as on the induction of nucleotide sequence variation can be investigated in further studies. 

### 2.4. Visualization of the Onion Root Tip following Treatment with Various Concentrations of HRE

Nitroblue tetrazolium (NBT) staining is a histochemical method that is used to measure concentrations of the superoxide anion [26], which is revealed by the development of dark blue coloration, and it has been used widely, as reported by Halliwell [27]. In previous studies, NBT staining has been performed using Arabidopsis (*Arabidopsis thaliana*) [28], mungbean (*Vigna radiate* L.) [29,30], and strawberry (*Fragaria* × *ananassa*) [31] to analyze the superoxide anion occurrence in plant organs such as seeds and leaves. 

The results of superoxide anion accumulation in the onion root are illustrated in Figure 4. After 3 days of treatment with various HRE concentrations, the tips of the onion root were stained with NBT, and a change in color was observed. NBT staining involves the use of a histochemical that is used as a probe for superoxide anion [26]. In the control group, the color of the onion root tips was deep blue (Figure 4A). As the HRE treatment concentration increased, the yellow area in the onion root tips expanded. After 0.3% HRE treatment, the color of onion root tips was completely yellow.

Figure 4B illustrates the percentage of the dead tissue area in the onion root. In the control root, the area of dead tissue was 0%; however, in the 0.1% and 0.2% HRE treatment roots, the ratio was 5% and 50%, respectively. This result differed significantly from the control (*p* = 0.01). In 0.3% HRE treatment root, the area of dead tissue was 80%, and it significantly differed from that of the control root (*p* = 0.001).

ROS are important signaling molecules for regulating root growth [32]. ROS are attributed with cell elongation through polysaccharide cleavage in cell walls [33]. The superoxide anion reacts with the nitric oxide radical to form peroxynitrite that quickly protonates into peroxynitrous acid, which adversely affects organic compounds [34]; however, this has also been attributed to increased cell expansion and proliferation. Furthermore, superoxide can increase growth in *A. thaliana* by 20%, and it accumulates in sites of growth in the roots of *Zea mays* [33]. ROS accumulation in the root tips of *A. thaliana* increases with the onset of growth and continues as the root elongates in the wild-type control groups [35]. 

Singh et al. [30] reported that the emergence of a radical is essential in seed germination and early axis growth. Considering that the occurrence of ROS is related to cell growth and division, HRE is presumed to inhibit cell division. In the 0.3% HRE treatment group, the root was thick and completely yellow in color. This indicates that 0.3% HRE completely inhibited the occurrence of superoxide anion, indicating that the root cells were almost dead. When referring to reports revealing that glucosinolate functions as a herbicide at high concentrations, it can be estimated that 0.3% HRE almost killed the onion root cells. According to Ciniglia et al. [36], ROS accumulation after allelochemical stress begins 3 h after treatment and peaks in the majority of samples after approximately 12 h, with a marked decrease in ROS after 24 h. In the current study, staining was performed 3 days after the HRE treatment; thus, the ROS generated after allelochemical stress were likely decreased. In addition, as the HRE concentration increased, the allelopathic effect was intensified and the stress on the cells increased; accordingly, the apoptosis rates increased, which eventually affected onion root growth.

### 2.5. Lettuce Root and Root Hair Measurement

Lettuce seeds were treated with the volatile compounds of HRE showed physiological changes including significant dose-dependent inhibition of root length and root hair length. (Figure 5A). In addition, the color of lettuce roots treated by 4% and 8% HRE were changed to light yellow, implying a consistent result with onion root assay above (Figure 4). A higher concentration including 4% and 8% of HRE would have significantly inhibited cell division of lettuce and induce apoptosis. In our recent study using lettuce leaves, AITC significantly induced lipid peroxidation in high AITC concentration in sealed Petri dish [37]. Song and Ku [37] also reported that AITC treatment significantly increased weight loss and the electrolyte leakage of lettuce, which indicates disrupted membrane integrity. The observed membrane damage in previous lettuce study [37] led to lipid peroxidation. Average lettuce root hair length decreased in a dose-dependent manner as HRE concentration was increased. Root hair length of control and 1%HRE lettuce were 1.30 mm, which is significantly shorter than 2% (0.76 mm), 4% (0.68 mm), and 8% (0.20 mm) of HRE (Figure 5B). For root length, control (21.9 mm), 1% HRE (22.6 mm), and 2% HRE (23.3 mm) were not significant with each other, although they were significant different compared to 4% HRE (17.0 mm) and 8% HRE (10.9 mm, Figure 5C). Eight percent HRE was significant shorter than all concentrations of HRE and control. The HRE treatment on lettuce was more effectively inhibit root hair length than root length, indicating that root hair is more venerable than main root to HRE. Root hair has significantly higher surface area than main root to absorb water and nutrients. In addition, root hair would have been exposed to HRE first due to the position. These factors probably make root hair more venerable to HRE. This phenomenon was clearly observed in previous study and displayed high-definition microscope images [19].

### 2.6. Metabolites Changes by HRE Treatment in Lettuce Seedlings

We compared metabolites among control or HRE treated lettuce spouts. We excluded 4% and 8% due to low yield because they were subjected with lethal dose of HRE. Serine, aspartate, beta-aminoisobutyric acid, glutamate, fructose, glucose, quinic acid, and myo-inositol were considerably lower in HRE 1% or 2% treatments (Figure 6). However, proline, valine, tyrosine, isoleucine, leucine, phenylalanine, and histidine were significantly lower than control. Previous study reported that the juglone treatment inhibited the growth of tobacco seedlings and increased reactive oxygen species content in tobacco roots with increased proline concentration [38]. Furthermore, plants pretreated with proline and then exposed to juglone showed attenuated toxic effects in roots, suggesting proline attenuate allelochemical juglone-induced stress in tobacco [38].

### 2.7. Metabolism Pathway Changes by HRE Treatment in Lettuce Seedlings

The most significantly changed metabolism was arginine and proline metabolism (Figure 7). Proline accumulation primarily occurs in response to drought but can also occur in many other stresses [39]. Glutamate is the key component of glutathione metabolism and is involved in defense against ROS. Previous study reported that AITC induced depletion of glutathione and the affected on inducing the detoxification enzyme such as GST in *Arabidopsis thaliana* [11]. Thus, the significantly changed glutathione metabolism also additionally explains that excessively high AITC concentration causes cellular damage by depleting glutathione, leading to cell death. Previous study reported that AITC treatment on lettuce affected energy metabolism by changing glucose and fructose levels [37]. This metabolic change related with starch and sucrose metabolism, which is energy metabolism. Tricarboxylic acid cycle (TCA) cycle was also significantly impacted by HRE treatment. Horseradish major volatile compounds were identified as AITC as well as phenethyl isothiocyanate (PEITC) [2]. Some metabolism changes may be synergistically influenced by PEITC along with AITC. Previously, pathway analysis revealed new metabolism insight to postharvest storage, drought stress, and AITC treated mature lettuce [37,40,41]. Tyrosine metabolism, isoquinoline alkaloid biosynthesis, and glyoxylate and dicarboxylate metabolism were also identified as significantly changed metabolism by pathway analysis. However, how these metabolisms relate with allelopathic effect by HRE aqueous extract requires further research. The allelopathic effect of HRE probably uses not a single mode of action, as the pathway analysis suggested. It is possible that HRE affects these metabolisms simultaneously (including unknown mechanism of isoquinoline alkaloid biosynthesis) or in a cascade sequence.

## 3. Materials and Methods

### 3.1. Horseradish Root Extract (HRE) Treatment and Onion Root Growth Rate Measurement

The onions (*Allium cepa* L.) used in this study were purchased from a local grocery. The old roots of the onion were removed using a scalpel (No. 10, Kai Medical, Tokyo, Japan), and the onions were placed in deionized water to induce new roots. After 3 days of growth, the onions with similar root growth rate were selected and were randomly divided into five groups.

Horseradish roots of accession IL706 [2] were freeze-dried and ground to a fine powder. Thereafter, the horseradish root powder was mixed with deionized water and filtered through a filter paper (Qualitative 102 Medium, Lab Nerd Scientific, St. Augustine, FL, USA) to prepare HRE at the final concentrations of 0.1%, 0.2%, and 0.3% (*w*/*v*).

The 0.1%, 0.2%, and 0.3% HRE preparations or deionized water (control) were added to a 266 mL transparent plastic cup until all onion roots were immersed (Figure 8). Five groups of onion bulb were plugged on plastic cups for 3 days, the cups and onions were covered with Parafilm to avoid the loss of the volatile compounds of HRE, and the length of the two longest roots of each onion bulb was measured every day.

### 3.2. Cell Cycle Analysis

After 3 days of HRE treatment, the onion root tips were collected for cell cycle analysis. Three root tips per onion were cut in five replications and incubated in 1 N HCl for 15 min at 60 °C. Thereafter, the root tips were washed with deionized water and stained in Schiff reagent for 15 min at room temperature (20–21°C). After staining, the onion root tips (1–2 mm) were again washed with deionized water and were then placed on a slide to analyze the cell cycle of the constituent cells under an optical microscope (BX53, Olympus, Tokyo, Japan) equipped with a digital camera (DP26, Olympus, Tokyo, Japan). The cells were examined for the various stages of the cell cycle, namely, prophase, metaphase, anaphase, and telophase, and the percentage of each mitotic phase was calculated.

### 3.3. Accumulation of Hydrogen Peroxide and Dead Area Measurement

Nitroblue tetrazolium chloride (NBT) assay was used to measure the accumulation of hydrogen peroxide in the onion root tips. Onion root samples (~1 cm) were placed in 1.5 mM NBT solution in 10 mM Tris-HCl for 8 min at room temperature and then washed with deionized water. The stained root samples were observed under a dissection microscope (Leica 125, Leica Microsystems, Wetzlar, Germany), and the image was captured. Dead area of onion root tips after NBT assay was measured using color threshold adjustment with ImageJ software (National Institutes of Health, Bethesda, MD, USA) in order to calculate the percentage of dead cells.

### 3.4. Horseradish Allelopatic Effect on Lettuce Growth

“Grand Rapids TBR” lettuce seeds (*Lactuca sativa*) were sown on 1.5% *w*/*v* agar on Fisherbrand square 100 mm by 100 mm polystyrene petri dishes (Fisher Scientific, Pittsburgh, PA, USA). Agar was prepared with agar agar powder (Landor Enterprises Inc., Williamsport, PA, USA) in deionized distilled water (1.5% *w*/*v*) by autoclave for 15 min at 121 °C. Twenty seeds were sown on each dish with radicles facing down. There were 4 treatment groups each consisting of 3 Petri dishes (*N* = 60 seeds). Dishes were treated with 1 mL of either deionized distilled water for control or 1%, 2%, 4%, or 8% HRE for treatment. The solution was carefully pipetted into the bottom of the dishes so that only the volatile compounds would interact with the seeds sown, on the upper third of the dishes. Dishes were sealed with Parafilm to keep the volatile compounds inside. Dishes were leaned against a wall with about a ~110° angle for support with a day/night photoperiod of 16 h d-1 at room temperature. Lettuce seeds were grown for 3 days and pictures were taken using a dissection microscope (Leica 125, Leica Microsystems, Wetzlar, Germany). Root length and root hair length of the samples were measured using ImageJ software (National Institutes of Health, Bethesda, MD, USA).

### 3.5. Primary Metabolite Extraction and Analysis

Primary metabolites were extracted following published protocols [40] with modifications of extraction solvent volume. Samples (10 mg) were weighed into 2 mL microcentrifuge tubes, followed by the addition of 80 μL of ribitol (10 mg/mL) as an internal standard, and extracted with 1.4 mL of methanol at 75 °C. After cooling, sample extracts were centrifuged at 15,000× *g* for 3 min, and 0.7 mL supernatant samples were transferred to new 2 mL microcentrifuge tubes. To fractionate polar compounds, 0.375 mL of cold chloroform (–20 °C) and 0.7 mL cold water (4 °C) were added. After vigorous mixing, the extracts were centrifuged at 15,000× *g* for 3 min, and 50 μL supernatant samples were transferred to 1.5 mL microcentrifuge tubes. The extracts were dried using a Vacufuge concentrator (Eppendorf, Thermo Fisher Scientific, Waltham, MA) with 10 μL of methanol to facilitate water evaporation. Dried extracts were derivatized with 50 μL methoxyamine hydrochloride (40 mg/mL in pyridine) for 90 min at 37 °C, then with 70 μL MSTFA + 1% TMCS at 37 °C for 30 min. Metabolites were analyzed using a GC-MS (Trace 1310 GC, Thermo Fisher Scientific, Waltham, MA, USA) coupled to an MS detector system (ISQ QD, Thermo Fisher Scientific, Waltham, MA, USA) and an autosampler (Triplus RSH, Thermo Fisher Scientific, Waltham, MA). A capillary column (Rxi-5Sil MS, Restek, Bellefonte, PA, USA; 30 m × 0.25 mm × 0.25 µm capillary column w/10 m Integra-Guard Column) was used to detect polar metabolites. After an initial temperature hold at 80 °C for 2 min, the oven temperature was increased to 330 °C at 15 °C/min and held for 5 min. Injector and detector temperatures were set at 250 °C and 250 °C, respectively. An aliquot of 1 μL was injected with the split ratio of 70:1. The helium carrier gas was kept at a constant flow rate of 1.2 mL/min. The mass spectrometer was operated in positive electron impact mode (EI) at 70.0 eV ionization energy at m/z 40–500 scan range. All data were normalized to ribitol, the internal standard in online platform MetaboAnalyst, and further statistical analysis was conducted after auto-scaling. Metabolite identification was based on standard compounds (STD) in comparison with the mass spectra in The National Institute of Standards and Technology (NIST) and retention time.

### 3.6. Statistical Analysis

Each treatment comprised five replications. Analysis of variance and *posthoc* test were performed using JMP Pro 12 (SAS Institute, Cary, NC, USA). Fisher’s least significant difference test was conducted for comparing the treatment group means at *p* ≤ 0.05. The results are presented as mean ± standard deviation. Multivariate and pathway analysis were conducted using MetaboAnalyst. For calculation of significant pathway among different pathways, the node importance values calculated from centrality measures are further normalized by the sum of the importance of the pathway.

## 4. Conclusions

As a result of this study, it was possible to observe allelopathy of HRE in onion root and lettuce germination assay. Allelopathy is a natural defense of plants against external threats; by studying allelopathy, it is possible to identify allelochemicals that may be used to produce growth regulators and natural herbicides for sustainable agriculture [42]. Allelopathy is due to secondary metabolites and is often identified accidentally in nature. The results of this study showed that onion root cells and lettuce seed germination assays are useful for investigating allelopathy. Metabolomics and cell cycle analysis were useful tools to elucidate mechanism of HRE allelopathy. From this study, we identified that a mode of action of AITC-induced allelopathy is associated with cell cycle arrest by AITC and several metabolisms including arginine and proline metabolism, glutathione metabolism, starch and sucrose metabolism, and other metabolism.

## Figures and Tables

**Figure 1 plants-10-01992-f001:**
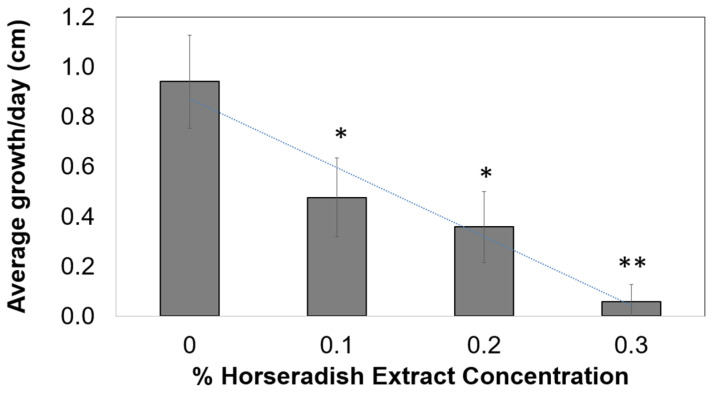
Average growth per day (cm) of onion root tips over 3 days during treatment with 0% control (without HRE) and solutions with 0.1%, 0.2%, and 0.3% concentrations of HRE. * and ** indicate significant difference from control by Student’s *t*-test at *p* = 0.05 and *p* = 0.01, respectively.

**Figure 2 plants-10-01992-f002:**
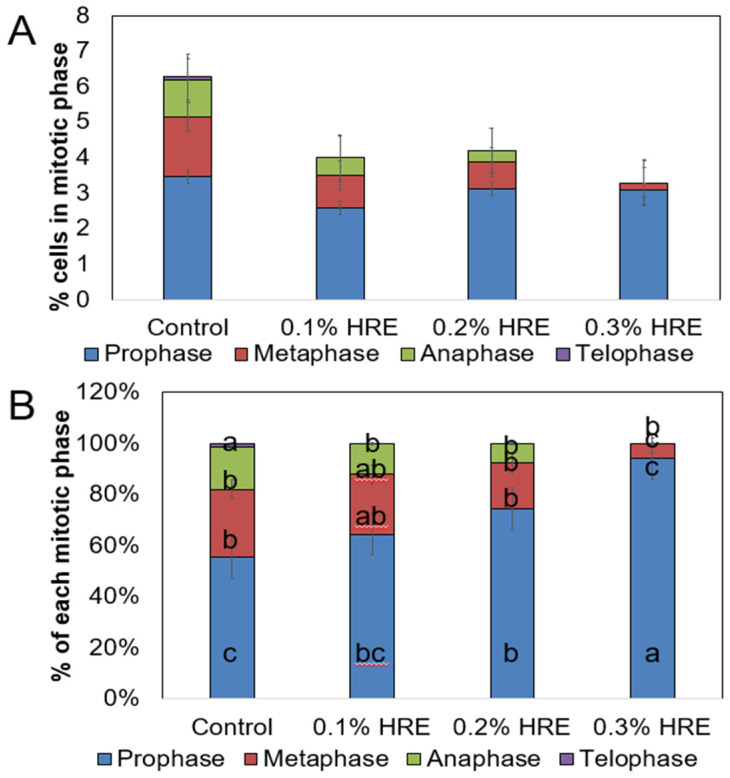
The percentage of cells in each mitotic phase was revealed after 3 days of treatment (**A**), and the percentage of each mitotic phase among the mitotic cells was also evaluated (**B**). HRE treatment was performed at 0% (control), 0.1%, 0.2%, and 0.3% concentrations. Different letters indicate significant difference among treatments by least significant difference test at *p* = 0.05.

**Figure 3 plants-10-01992-f003:**
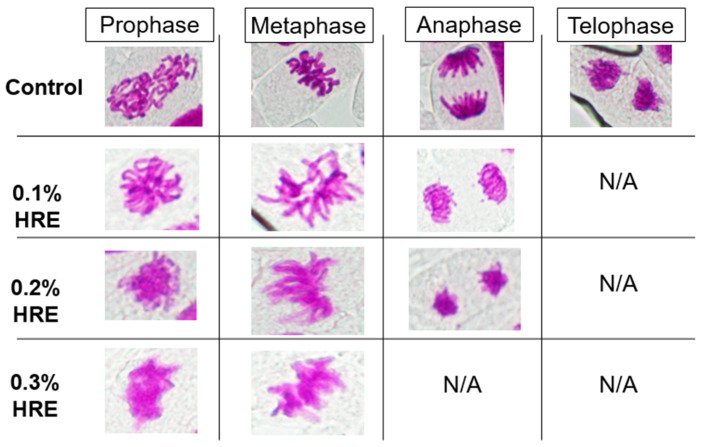
Representative images of mitotic stages after the treatment of the onion root tip with various concentrations HRE. Onion root tips were dyed to observe the chromosomal status of each cell at HRE concentrations of 0% (control) and 0.1%, 0.2%, and 0.3%. We distinguished the mitotic stages of the cells into prophase, metaphase, anaphase, and telophase. N/A means not applicable. HRE, horseradish extract.

**Figure 4 plants-10-01992-f004:**
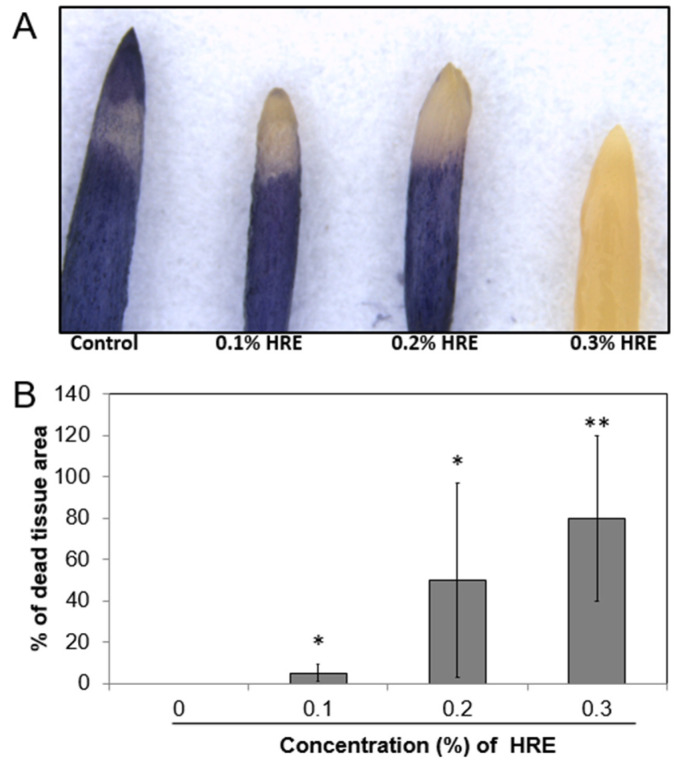
Nitroblue tetrazolium staining (ROS accumulation) of onion root tips (**A**) and dead tissue area after 3 days of treatment with various HRE concentrations (**B**). * and ** indicate significant difference from control by Student’s *t*-test at *p* = 0.01 and *p* = 0.001, respectively. HRE, horseradish extract.

**Figure 5 plants-10-01992-f005:**
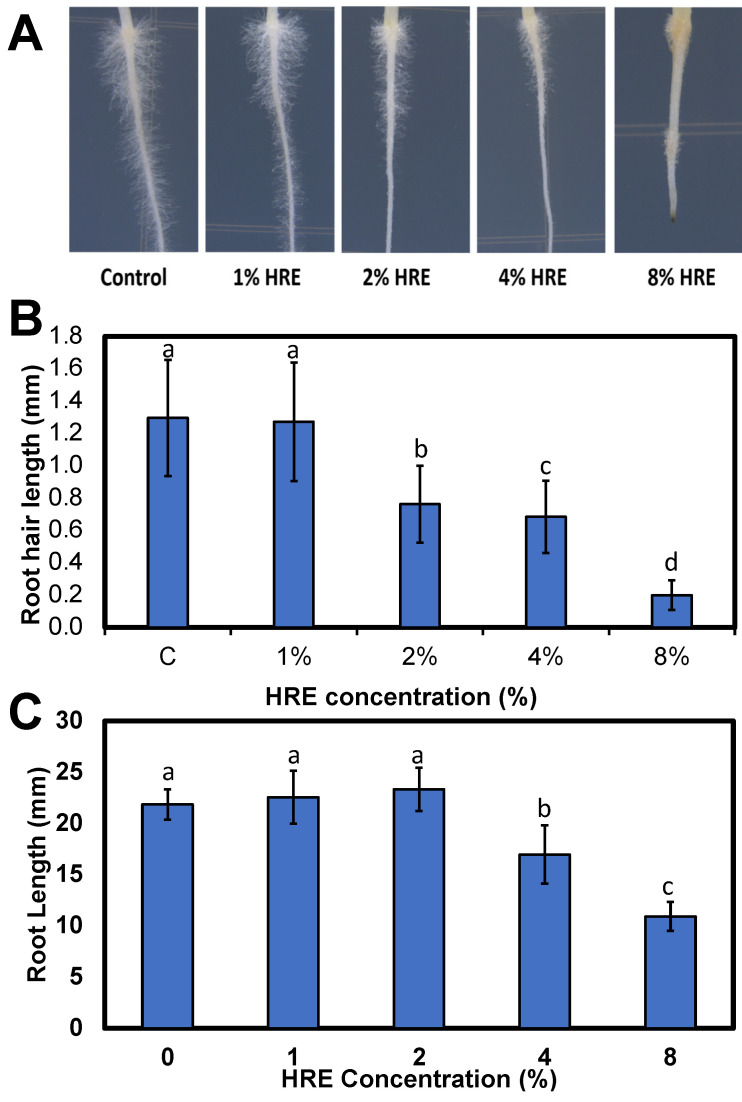
Lettuce root photo (**A**), root hair length (**B**), and root length (**C**) after applying using various concentrations of horseradish extract (0, 1, 2, 4, and 8% aqueous extract) for 3 days. “Grand Rapids TBR” lettuce seeds (*Lactuca sativa*) were sown on 1.5% *w*/*v* agar on square petri dishes. There were 4 treatment groups each consisting of 3 Petri dishes (*N* = 60 seeds). Petri dishes were sealed with Parafilm to keep the volatile compounds after applying with 1 mL of either deionized distilled water for control or 1%, 2%, 4%, or 8% HRE for treatment. Different lowercase letters indicate significant difference at *p* < 0.05 by least significant difference.

**Figure 6 plants-10-01992-f006:**
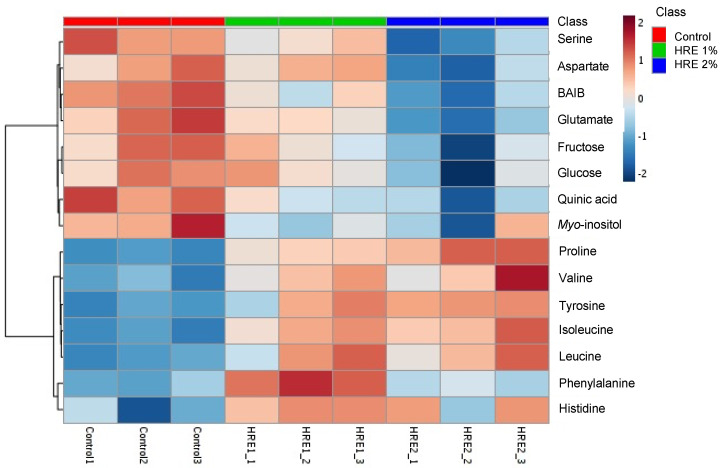
Heatmap of significantly changed lettuce metabolites after 3 days of water, 1%, or 2% horseradish aqueous extract (HRE) treatments. BAIB = beta-aminoisobutyric acid.

**Figure 7 plants-10-01992-f007:**
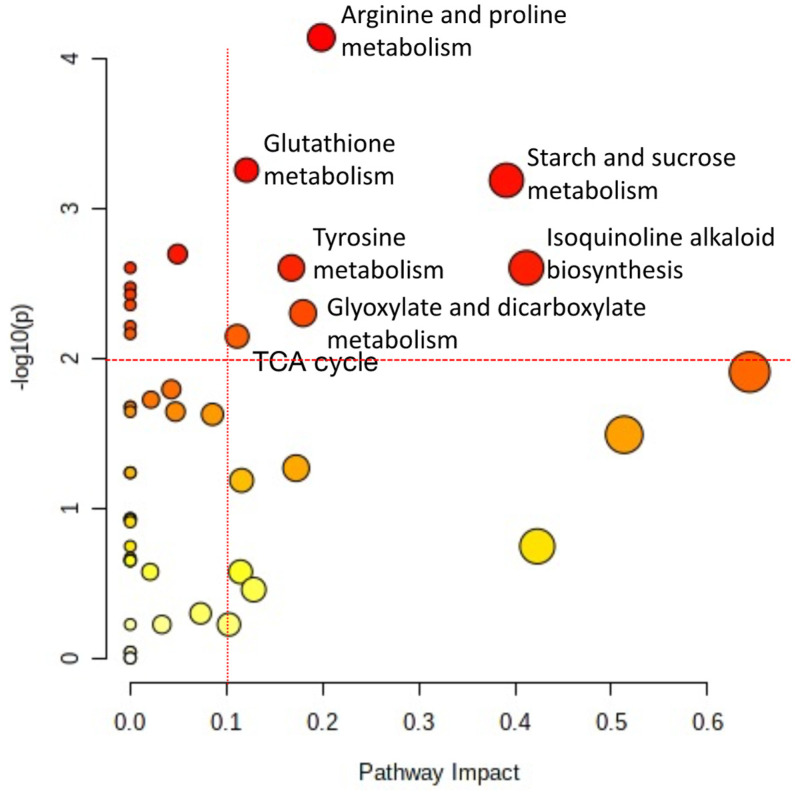
Significantly changed lettuce metabolism after 3 days of water, 1% and 2% horseradish water extract (HRE) treatments based on MetaboAnalyst pathway analysis. X-axis displays the pathway impact values that were estimated by topology analysis, and the circle sizes correspond to the pathway impact scores; Y-axis represents the -log(p) values that were calculated by over-representation analysis, and the circle colors correspond to *p* values (darker colors indicate more significant changes in metabolites in the corresponding pathway). The represented pathway names that were most significantly changed are characterized by both a high -log(p) value (>2) and high impact value (0.1).

**Figure 8 plants-10-01992-f008:**
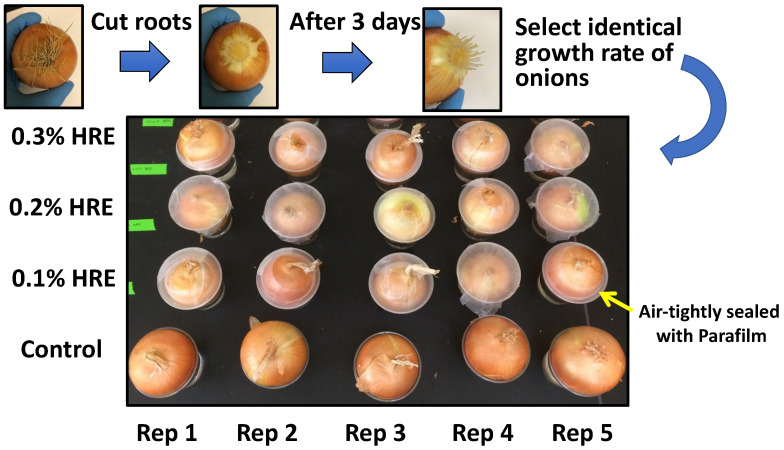
Onion root growth rate (per day) measurement using various concentrations of horseradish extract (0%, 0.1%, 0.2%, and 0.3% aqueous extract). Identical growth rate of onion roots was selected and used for this allelopathic assay after cutting onion roots and regrowth 3 days.

## Data Availability

All data are included in the present study.

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
