# Peer review of "Metabolomics and Physiological Approach to Understand Allelopathic Effect of Horseradish Extract on Onion Root and Lettuce Seed as Model Organism"

_plants, 2021, doi:10.3390/plants10101992_

Round 1
Reviewer 1 Report
All shortcuts should be explained before first mention in manuscript, e.g. AITC, ABA, GST…
Line 74: „Our results confirmed the allelopathic effect of HRE on onion root” – it suggests that the allelopathic effect of HRE on onion root was already previously investigated
Line 90: - unfinished statement: „In contrast to the result that glucosinolate can inhibit the germination of other plants,”
Line 106: correct „di-viding”
Line 213: „treated with the volatile compounds of HRE” – seeds were treated with all components of HRE (not only volatile)
Line 274: „Tyrosine metabolism, Isoquinoline Alkaloid biosynthesis, and Glyoxylate and Dicarboxylate metabolism were also identified” - How was metabolism identified?
Fig. 7: On what basis was this Figure prepared? No mention in experimental section regarding analysis of alkaloids, starch etc…
Line 299: It is not clear how was the extract concertation established. Extract was evaporated to dryness? Or powder was accurately weighted and suspended in water?
Fig. 8. Legend is not precise: „Onion root growth rate measurement”? Moreover, Fig. 8 is not cited in the text.
Conclusion section should be rewritten because it is too general (see line 380-383).
Author Response
Dear Reviewer,
Thank you so much for your detail comments.
It helped a lot to improve our manuscript.
We revised as much as we can based on your comments.
Best,
Kang Mo Ku

Reviewer 2 Report
Your paper addresses a topical scientific field - modern agriculture, which involves, among other things, the use of vegetal herbicides and growth regulators.
The abstract is a objective representation of the article.
The research is well motivated in Introduction.
The metabolomic and allelopathic researches are well conducted.
The results are clearly presented and the discutions are relevant.
The bibliography is exhaustive.
In my opinion, some additions and revisions are needed, as follow:
- Line 25: ,,Allium cepa” instead ,,Allium cepa L.” (italic font)
- ,,Lactuca sativa . may be added to keywords (it is only a suggestion)
- Explain the abbreviations for the first use in the text (even if they are common), for example: ABA (line 51), GST (line 54), GSH (line 58), ROS (line 69), TCA (line 269) s.a.
- Lines 90-91: revise the text
- 2: you must complete the legend (what does it mean ,,a”, ,,b”, ,,c” ?)
- Line 200: ,, Ciniglia et al.” instead,, Ciniglia, et al.”
- Fig 7: ,, TCA cycle” instead,, TCA cycle)”
- Line 288: ,, (0.1)” instead ,, (0.1))”
- The bibliography must be corrected, for example:
[3]: Shofran, B.G., Purrington S.T., Breidt F., and Fleming H.P…….” instead ,,SHOFRAN, B.G., S.T. PURRINGTON, F. BREIDT, and H.P. FLEMING…….”
[8], [11], [12], [17], [23], [28]: correct the authors (see Plants Microsoft Word template file: Author 1, A.B.; Author 2, C.D…..)
Author Response
Dear Reviewer,
Thank you so much for your detail comments.
It helped a lot to improve our manuscript.
Best,
Kang Mo Ku
